# Vaccine Hesitancy among Italian Patients Recovered from COVID-19 Infection towards Influenza and Sars-Cov-2 Vaccination

**DOI:** 10.3390/vaccines9020172

**Published:** 2021-02-18

**Authors:** Valentina Gerussi, Maddalena Peghin, Alvisa Palese, Valentina Bressan, Erica Visintini, Giulia Bontempo, Elena Graziano, Maria De Martino, Miriam Isola, Carlo Tascini

**Affiliations:** 1Infectious Diseases Division, Santa Maria Misericordia University Hospital, 33100 Udine, Italy; valentina.gerussi@gmail.com (V.G.); bontempo.giulia@gmail.com (G.B.); elenagraziano22@gmail.com (E.G.); c.tascini@gmail.com (C.T.); 2Department of Medical Sciences, School of Nursing, University of Udine, 33100 Udine, Italy; alvisa.palese@uniud.it (A.P.); valentina.bressan@uniud.it (V.B.); visintini.erica001@spes.uniud.it (E.V.); 3Department of Medicine, University of Udine, 33100 Udine, Italy; maria.demartino@uniud.it (M.D.M.); miriam.isola@uniud.it (M.I.)

**Keywords:** COVID-19, vaccination, influenza, pandemic, vaccine hesitancy, SARS-CoV-2

## Abstract

We aimed to assess the attitude towards influenza and severe acute respiratory syndrome coronavirus 2 (SARS-CoV-2) vaccinations among coronavirus disease 2019 (COVID-19) recovered patients. We performed a cross-sectional study consisting of a standardized telephone interview carried out between September and November 2020 targeting a cohort of adult in- and out-patients that had recovered from COVID-19 after the first wave (March–May 2020) at Udine Hospital (Italy). Overall, 599 people participated (320 female, median age 53 years) and most had experienced an acute COVID-19 with mild illness (409, 68.3%). The majority were hesitant or undecided towards influenza (327, 54.6%) and SARS-CoV-2 (353, 59.2%) vaccines. Older age, public work exposure, and previous 2019 flu shots were the main factors associated with a positive attitude toward both vaccinations (*p* < 0.05). Being hospitalized during the acute COVID-19 phase was associated with the willingness to get a flu shot (94/272, 34.5%) but not SARS-CoV-2 vaccine (70/244, 28.7%). Vaccine hesitancy is diffuse and multifactorial also among COVID-19 recovered.

## 1. Introduction

The coronavirus disease 2019 (COVID-19) pandemic is a worldwide threat with Italy being the first country in Europe to be heavily hit by the virus. Due to the huge health, social and economic impact of the ongoing COVID-19 pandemic, the convergence of a simultaneous seasonal influenza epidemic is of great concern during the 2020–2021 winter season. As long as there is no effective and safe vaccine to protect individuals at risk of severe COVID-19, non-pharmaceutical interventions (NPIs) have been documented as the most effective public health strategy against COVID-19 capable to significantly contain the pandemic [1]. Governments worldwide have implemented varying degrees of restrictions to slow the spread of the severe acute respiratory syndrome coronavirus 2 (SARS-CoV-2) with different epidemiological responses [2]. However, in several regions, such as that where this study has been performed (Friuli Venezia Giulia, Udine), NPIs have been documented to be insufficient to control the spreading of respiratory viral infections [3]. Therefore, flu vaccination strategies during the COVID-19 pandemic have been underlined to be crucial in order to mitigate disease-specific burden as well as to reduce the strain on healthcare systems’ demand [4]. On the other side, due to the urgency imposed by the pandemic, the development of a vaccine against COVID-19 has accelerated at an unprecedented pace. Over 50 international clinical trials are ongoing and vaccine candidates have been recently approved for emergency use [5]. 

However, despite the documented safety and effectiveness of the immunization practice, vaccine hesitancy has become an emerging global issue and it was identified by the World Health Organization (WHO) as one of the top ten threats to global health in 2019 [6]. Yet, the determinants of vaccine tendency are still poorly understood also due to their multidimensional features [7]. Despite the well-established Italian immunization policies, alarming reductions in vaccine coverage have been recently observed [8]. In this specific situation, the streamlined and speedy development of a vaccine against the SARS-CoV-2 adds up as a further causative feature of public concern hence compromising acceptance [9].

To our best knowledge, no studies to date have investigated the vaccine hesitancy towards flu and COVID-19 among patients who had been infected with SARS-CoV-2 during the first pandemic wave. Filling in this gap might contribute towards describing the phenomenon as well as to the design specific public strategies aimed at contrasting hesitancy.

## 2. Materials and Methods

We carried out a cross-sectional study in 2020 at the Academic Hospital Udine (Italy), a tertiary-care teaching hospital (1000 beds) that is also a referral regional center for COVID-19 attending a population of approximately 530,000 inhabitants. 

### 2.1. Participants

The target population was a cohort of adult in- and out-patients visiting the Infectious Disease Department from 1 March 2020—the day when the first COVID-19 case was found—to 30 May 2020. There sample comprised eligible patients >18 years, with suspected or confirmed COVID-19 cases, and who were willing to participate in a telephonic interview performed after 6 months of the infection onset.

### 2.2. Primary Outcome and Associate Variables 

The primary aim of the study was to assess patient’s tendency towards the flu vaccine and a putative COVID-19 vaccine. The secondary aim was to identify associated factors of hesitancy.

Vaccine tendency was categorized in three groups: “likely” as those who were prone to vaccinate, “unlikely”, and “undecided” according to participants who answered, respectively, “Yes”, “No” and “I don’t know” to the question: “Will you take influenza/SARS-CoV-2 vaccine?”. Participants were free to justify their SARS-CoV-2 vaccine hesitancy; then, answers were grouped into six categories relying on existing classifications of attitudes towards vaccines [10] (Box 1). 

A confirmed COVID-19 case was defined as a patient with a positive nucleic acid amplification test (NAAT) for SARS-CoV-2 in respiratory tract specimens; a suspected COVID-19 case was defined as a patient suspected of having COVID-19 with negative SARS-CoV-2 NAAT, but showing characteristic laboratory or imaging findings [11]. For asymptomatic patients, the onset of symptoms was considered as the first day of NAAT positivity. 

Moreover, based on COVID-19 disease severity scale, patients were classified into five groups: asymptomatic; mild disease (without pneumonia); moderate disease (pneumonia); severe disease (severe pneumonia); critical disease, including acute respiratory distress syndrome (ARDS), sepsis and/or septic shock [12]. Moreover, patients were divided into three groups: intensive care unit (ICU) group, if they were cared for in an ICU; hospital ward group, if they were admitted to a hospital ward (Infectious Disease Department, Emergency Department, Pneumology); outpatient group, if they were never hospitalized and included both symptomatic and asymptomatic patients. Furthermore, in all of the patients included, we collected some baseline characteristics (e.g., age, gender, country of birth, baseline comorbidities, alcohol and smoking habits, work, previous flu vaccination) that have been associated with vaccine hesitancy [6].

### 2.3. Data Collection

All eligible patients were contacted by phone by trained nurses between September and November 2020. The responses were recorded using a standardized interview guide (Box 1) piloted for feasibility and understandability in 10 patients. No changes were required during the pilot phase. Participants were free to answer with their own words and interviews ranged from 10 to 30 minutes. Clinical data collected at the hospital admission and during the follow-ups were extracted from the General Hospital and Microbiology databases, using a standardized protocol. 

### 2.4. Ethical Issues

All procedures were in accordance with the ethical standards of the University of Udine and Azienda Sanitaria Universitaria Integrata di Udine (CEUR-2020-OS- 219/CEUR-2020-OS-205) and with the 1964 Helsinki Declaration and its later amendments or comparable ethical standards. Verbal informed consent was obtained from all subjects before being contacted for the interview.

Box 1Standardized questionnaire on attitude towards vaccine and SARS-CoV-2 vaccine.Did you take flu vaccine last year 2019? (Yes/No)Will you take flu vaccine
this year 2020? (Yes/No/I don’t know) §
If it was available,
would you take SARS-CoV-2 vaccine? (Yes/No/I don’t know) §
If not, why?
-I am concerned about the
safety and/or the side effects
-I am concerned because I
don't think the vaccine will be effective-I don’t think I will need
the vaccine due to previous infection and/or health status-I am against vaccines in general-I can’t take any vaccine
because of previous vaccine reactions
-I don’t know


**§** No = hesitancy or unlikely; I don’t know = undecided.

### 2.5. Statistical Analysis

Descriptive statistics included frequency analyses (percentages) for categorical variables and mean (standard deviation; SD), median and interquartile range (IQR) for quantitative variables. Data were tested for normal distribution using the Shapiro–Wilk test. 

Patients were stratified by age (intervals 18–44, 45–64, >65 years old) and by other demographic and clinical characteristics. In order to explore vaccine hesitancy, the Chi-square (χ2) test or Fisher test were used to compare categorical variables among groups, as appropriate. One-way ANOVA or Kruskal–Wallis test were used to compare continuous variables among groups, as appropriate. Analyses were performed using STATA 16. A *p*-value ≤ 0.05 was considered significant.

## 3. Results

### 3.1. Baseline Characteristics of COVID-19 Population

Overall, 1067 COVID-19 diagnoses were performed in our hospital during the study period. After excluding 240 patients for refusal to participate in the research, 138 nursing home or long-term facility residents not capable to answer due to cognitive decline, nine lost-to-follow-up and 81 deaths, 599 patients were included.

Mean age was 53 years (SD = 15.8; range 18–94), 320 (53.4%) were women and 524 (91.5%) were native Italian. Health care workers (HCWs) accounted for 23.9% of the respondents and 18.0% worked in contact with public (Table 1).

Most patients (312/593, 52.6%) reported at least one chronic medical condition and the most frequent underlying diseases included hypertension (135/586, 23.0%) and diabetes (33/593, 5.6%) (Table 2).

During acute COVID-19 disease, the majority of patients (541/596, 90.8%) experienced symptomatic COVID-19 infection and most of them were classified as mild (409/541, 75.6%) and moderate (93/541, 17.2%) cases.

Among all participants, 134 (22.4%) were hospitalized and 23 (3.8%) were admitted in an ICU (Table 3).

### 3.2. Attitude towards Influenza Vaccination

Overall, approximately one out of four patients (157/599, 26.3%) reported receipt of influenza immunization in 2019 and about one in two (272/599, 45.4%) reported willingness to undertake immunization in 2020. Among the remaining recovered COVID-19 patients, 131 (21.9%) were undecided and 196 (32.7%) were hesitant to accept influenza vaccination. 

As reported in Table 1 and in Table 2, several factors showed an association with the likelihood of acceptance of influenza vaccination as: age (70.4% > 65 years vs. 24.9% 18–44 years; *p*-value < 0.001), chronic diseases (54.2% vs. 36.6%; *p*-value < 0.001) and an increasing number of comorbidities (64.7% vs. 35.8%; *p*-value < 0.001) at the baseline. Influenza vaccine tendency was significantly associated with occupational exposure risk, based on activities with or without contacts with the public (42.9% vs. 37.5%; *p*-value < 0.001). Among HCWs, only 31/133 (23.5%) reported to have taken the flu shot in 2019, while a higher proportion (60/133, 45.1%) were likely to take it in 2020. Patients who underwent influenza immunization in 2019 were at increasing likelihood of accepting influenza vaccine immunization as compared to patients who had not been vaccinated in the previous year (*p*-value < 0.001). 

As reported in Table 3, influenza vaccine hesitancy was significantly lower in COVID-19 patients who experienced hospital admission (*p*-value 0.001). However, the duration of neither symptoms nor hospital stay was associated with attitude toward flu vaccination, as well as with regard to the severity of the COVID-19 disease.

### 3.3. Attitude toward SARS-CoV-2 Vaccine 

Overall, most patients showed to be either undecided (204/597, 34.2%) or reluctant (149/597, 24.9%) to receive SARS-CoV-2 vaccine, without reporting the reasons (253/597, 42.4%). As described in Table 4, among patients who justified their choice, most (31/90) considered themselves not to need the vaccination according to their young age, health status or believed immunization due to previous COVID-19 disease; others manifested doubts on vaccine efficacy (28/90), fear (21/90), no-vax attitudes (18/90) and two referred previous vaccine adverse reactions.

As reported in Table 1, Table 2, Table 3, people > 65 years, those who previously received influenza immunization and those working in contact with the public seemed to be more prone to undertake the SARS-CoV-2 vaccination (51.3% vs. 41.5%, *p*-value < 0.015; 53.2% vs. 36.4%, *p*-value < 0.001; 40.2% vs. 37.5%, *p*-value 0.036, respectively). Gender appeared to be a predictor of SARS-CoV-2 vaccine tendency with more confidence in men and more indecision in women (45.7% vs. 36.7%, p-value < 0.050, respectively). 

No significant association emerged between SARS-CoV-2 vaccine hesitancy and clinical baseline characteristic. Finally, COVID-19 disease severity and hospitalization were not associated with the likelihood of acceptance of SARS-CoV-2 vaccination (*p*-value 0.486 and *p*-value 0.800, respectively).

## 4. Discussion

Hesitancy towards flu and SARS-CoV-2 vaccine among COVID-19 recovered patients and associated factors after six months of their infection were the primary and the secondary aims of the study. Understanding hesitancy prevalence and its determinants might support in shaping specific interventions aimed at overcoming modifiable barriers [13,14,15]. Moreover, the simultaneous presence of the SARS-CoV-2 pandemic and the seasonal influenza epidemic has been reported as a great concern at the beginning of the 2020–2021 winter season. Measures meant to control the SARS-CoV-2 pandemic have shown wide-ranging implications also on influenza and most other respiratory diseases in the Southern and Northern Hemisphere [16,17]: in this context, we investigated hesitancy regarding both vaccinations. 

At the overall level, less than 50% of the COVID-19 survivors reported a positive attitude toward both vaccinations while a substantial proportion of them were hesitant or undecided, in line with recent surveys on general population [18]. Age, work exposure and previous 2019 flu shot were the main associated factors increasing attitude toward both vaccinations, but surprisingly the hospitalization due to the infection was associated with willingness to get flu shot but not SARS-CoV-2 vaccine. 

Specifically, one out of three patients was undecided regarding the SARS-CoV-2 vaccine. This might be explained by (a) their negative recent disease experience, still burdening them; (b) the lack of clear public information regarding the SARS-CoV-2 vaccine; (c) the belief of being immune, as documented in surveys on vaccinal tendency among people after H1N1 [19,20]. In relation to this, at the beginning of the pandemic, immunity to COVID-19 was poorly known as well as data regarding reinfection. Our participants have been already infected, and they might consider themselves protected, a finding at merit of consideration given that according to available guidelines, vaccination should be offered to individuals regardless of prior SARS-CoV-2 infection [21]. Moreover, during the first wave, mortality and severity of the disease were associated with age and chronic diseases. We involved mainly <65 years, and without comorbidities, thus increasing the likelihood of perceiving low risk of getting a severe illness [22]. However, among those undecided as well as among hesitant patients towards SARS-CoV-2 vaccine, the majority reported no arguments, and this might reflect the uncertainty generated by the COVID-19 outbreak. 

Among the patient-reported reasons, worry about effectiveness and health risks regarding safety and potential side effects of the SARS-CoV-2 vaccine were the mostly narrated, in line with available studies [14]. On the one hand, this kind of concern was—to some extent—valid at the time of our survey as there were various types of vaccines under development with lack of public data. On the other hand, trust is a well-known determinant of vaccine attitude that might have been threatened by fake news often spread through social media [23]. In an era where most of the information comes from collective events—rather than from the physician’s recommendations—it becomes crucial to improve communication and emphasize collective—rather than personal—responsibility [24,25,26].

Nevertheless, findings underlined an improvement in flu vaccination attitude since almost one out of four participants had taken the flu shot in 2019 and half of the total was supposed to take it in 2020. The data seem to reflect the influenza coverage rate that did not reach the minimum recommended threshold of 75% and is far from the optimum target of 95% [27]. However, the increased attitude toward flu vaccination might be attributed to (a) the influenza immunization campaign 2020 that was longer and largest as compared to that conducted previously [28], but also to (b) the improved awareness of all respiratory viral illnesses generated by the COVID-19 individual and collective experience [29]. 

Alongside these reasons of hesitancy that remain difficult to clarify, other associated socio-demographic and clinical factors emerged. 

An age-related association with vaccine tendency has emerged towards both vaccines: older people were more willing to take the flu vaccine compared to younger participants. These results are in line with large-scale winter vaccination campaigns being mostly directed to people over 65 [27,30]. Age was associated also with COVID-19 vaccine as reported in previous studies performed in the US and France [31,32]. A greater perceived risk of getting infected and of developing a severe disease has been documented among older people [18].

Regarding gender, while no associations with flu vaccine attitude emerged, more males reported positive attitudes towards SARS-CoV-2 vaccine, which could be attributed to the higher incidence of severe disease among them as well as to a male propensity for pursuing behaviors that are felt as risk-taking [33]. 

A role in vaccine attitude is played also by the occupational exposure risk. The majority of participants willing to be immunized against flu and SARS-CoV-2 were in contact with the public at work. As previously documented, individuals at risk of disease are generally more prone to receive vaccination [20]. However, the vaccine tendency rate remains low among HCWs, despite their personal and collective experience [34]. Specifically, we found that around half of the HCWs were still refusing the influenza vaccination and they were skeptical regarding a SARS-CoV-2 vaccine as also reported recently [35]. Given their role in educating patients, the rate of hesitancy among HCWs should be considered with care while designing ad-hoc interventions aimed at improving vaccination adherence. 

Patients receiving influenza immunization in 2019 were more likely to accept both vaccines and this seems to confirm the theory of “tendency to persistence” with vaccination in the prior year being one of the most important predictors of vaccination [36]; moreover, this can also be interpreted as the presence of a certain degree of vaccine-likelihood innate in people [37].

Vulnerable groups are less hesitant to vaccinations and their frequent access to the hospital environment may favor contact with physician information and sensibilization campaigns [30,38,39,40]. In addition, in our study, the presence and the number of comorbidities positively influenced decision-making regarding flu shots but not regarding the SARS-CoV-2 vaccine. The SARS-CoV-2 hesitancy can again be associated with the time when the survey was carried and the dominant atmosphere of insecurity and lack of confidence towards COVID-19 prevention.

Perceptions and previous COVID-19 illness influenced the tendency towards flu but not towards the SARS-CoV-2 vaccine in our study. Moreover, according to the findings, hesitancy was lower in those who experienced critical COVID-19 illness as compared to those who experienced milder disease. No studies are available on COVID-19 disease severity and vaccine tendency [41], therefore, limiting any comparison. Indeed, the fear of feeling shortness of breath and respiratory symptoms might function as prevention incentive, as reported for flu vaccine [42]. 

Despite being methodologically in line with recent surveys performed on general population in Italy [43], our study has several limitations. Our interviews were performed at the end of the first wave. Moreover, interviews were performed at a time when the COVID-19 vaccination was not yet available; recent approval and information campaigns on vaccinations might have changed participants’ tendencies. Furthermore, reporting the willingness to be vaccinated might express a socially desirable answer, thus not necessarily a predictor of getting a vaccination. In addition, we focused only on COVID-19 patients and we did not consider a control group without previous SARS-CoV-2 infection. The lockdown experience as well as the changes in lifestyle habits, pandemic insecurity, work loss, poverty, ethnicity or education, have not been investigated. 

## 5. Conclusions

A pandemic is a community experience exerting a major impact on all citizens and requiring a collective response. However, vaccine hesitancy remains insidious and multifactorial even among COVID-19 survivors, since most recovered patients showed to be refusing or doubtful towards both flu and SARS-CoV-2 vaccinations. In our study, a significant increase in acceptance to both vaccinations is associated with older age, occupational exposure risk and previous experience of influenza vaccination. Nevertheless, the degree of severity of recent acute COVID-19 seems to have a favorable repercussion on the propensity towards influenza vaccination but not SARS-CoV-2 vaccination. Therefore, younger citizens, those working without contacts with the public, and those who have shown hesitancy towards flu vaccination, required tailored educational interventions. However, further studies and planned interventions are warranted in order to understand the complex and interplaying factors that influence vaccination decisions and to improve vaccine coverage, particularly during the COVID-19 pandemic.

## Figures and Tables

**Table 1 vaccines-09-00172-t001:** Demographic characteristics of participants according to their attitudes towards vaccination.

	Influenza Vaccine 2020				SARS-CoV-2 Vaccine 2020			
	Unlikely	Likely	Undecided	*p*-Value	Unlikely	Likely	Undecided	*p*-Value
	*n* (%)	*n* (%)	*n* (%)		*n* (%)	*n (*%)	*n* (%)	
Gender				0.825				0.049
Male	88 (31.5)	130 (46.6)	61 (21.9)		59 (21.2)	127 (45.7)	92 (33.1)	
Female	108 (33.8)	142 (44.4)	70 (21.9)		90 (28.2)	117 (36.7)	112 (35.1)	
Age				<0.001				0.015
18–44	81 (45.8)	44 (24.9)	52 (29.4)		45 (25.6)	73 (41.5)	58 (32.9)	
45–64	91 (33.7)	121 (44.8)	58 (21.5)		77 (28.6)	93 (34.6)	99 (36.8)	
>65	24 (15.8)	107 (70.4)	21 (13.8)		27 (17.8)	78 (51.3)	47 (30.9)	
Nationality				0.852				0.593
Italian	170 (32.4)	243 (46.4)	111 (21.2)		127 (24.3)	213 (40.8)	182 (34.9)	
European	17 (38.6)	17 (38.6)	10 (22.7)		15 (34.1)	15 (34.1)	14 (31.8)	
Extra European	2 (40.0)	2 (40.0)	1 (20.0)		1 (20.0)	3 (60.0)	1 (20.0)	
Occupation				<0.001				0.036
Workers, not with public	62 (40.8)	57 (37.5)	33 (21.7)		44 (28.9)	57 (37.5)	51 (33.5)	
Workers, with public	38 (38.0)	40 (40.0)	22 (22.0)		27 (27.0)	40 (40.0)	33 (33.0)	
Health care workers	44 (33.1)	60 (45.1)	29 (21.8)		31 (23.3)	54 (40.6)	48 (36.1)	
Other	21 (32.8)	26 (40.6)	17 (26.6)		12 (19.3)	18 (29.0)	32 (51.6)	
Retired	18 (16.8)	73 (68.2)	16 (14.9)		24 (22.4)	56 (52.3)	27 (25.2)	
Smoking habits				0.860				0.924
Non-smoker	125 (32.1)	182 (46.8)	82 (21.1)		97 (25.0)	158 (40.7)	133 (34.3)	
Current smoker	29 (35.4)	33 (40.2)	20 (24.4)		23 (28.05)	33 (40.2)	26 (31.7)	
Ex-smoker	40 (31.7)	57 (45.2)	29 (23.0)		28 (22.4)	53 (42.4)	44 (35.2)	
Alcohol consumption				0.007				0.154
Non-drinker	87 (29.1)	139 (46.5)	73 (24.4)		69 (23.2)	120 (40.4)	108 (36.4)	
Occasional drinker	64 (37.2)	65 (37.8)	43 (25)		49 (28.5)	63 (36.3)	60 (34.9)	
Daily drinker	42 (34.4)	66 (54.1)	14 (11.5)		28 (22.9)	60 (49.2)	34 (27.9)	
Alcohol abuse	1 (33.3)	2 (66.7)	0 (0.0)		2 (66.7)	2 (33.3)	0 (0.0)	
Flu shot 2019				<0.001				<0.001
No	193 (43.8)	133 (30.2)	115 (26.1)		128 (29.1)	160 (36.4)	152 (34.6)	
Yes	2 (1.3)	139 (88.5)	16 (10.2)		21 (13.5)	83 (53.2)	52 (33.3)	

**Table 2 vaccines-09-00172-t002:** Clinical characteristics of patients.

Heading	Influenza Vaccine 2020				SARS-CoV-2 Vaccine 2020			
	Unlikely	Likely	Undecided	*p*-Value	Unlikely	Likely	Undecided	*p*-Value
	*n* (%)	*n* (%)	*n* (%)		*n* (%)	*n* (%)	*n* (%)	
Hypertension				<0.001				0.078
No	169 (37.5)	180 (39.9)	102 (22.6)		119 (26.5)	172 (38.3)	158 (35.2)	
Yes	22 (16.3)	86 (63.7)	27 (20.0)		27 (20.0)	66 (49.9)	42 (31.1)	
COPD				0.714				0.360
No	184 (32.2)	261 (45.6)	127 (22.2)		139 (24.4)	236 (41.4)	195 (34.2)	
Yes	8 (38.1)	10 (47.6)	3 (14.3)		8 (38.1)	7 (33.3)	6 (28.6)	
Diabetes				0.516				0.215
No	185 (33.0)	254 (45.4)	121 (21.6)		135 (24.2)	233 (41.8)	190 (34.0)	
Yes (No complications)	6 (20.0)	16 (53.3)	8 (26.7)		11 (36.7)	8 (26.7)	11 (36.7)	
Yes (with complications)	1 (33.3)	1 (33.3)	1 (33.3)		1 (33.3)	2 (66.7)	0 (0.0)	
Chronic diseases				<0.001				0.523
No	114 (40.6)	103 (36.6)	64 (22.8)		74 (26.5)	116 (41.6)	89 (31.9)	
Yes	78 (25.0)	169 (54.2)	65 (20.8)		73 (23.4)	127 (40.7)	112 (35.9)	
Chronic diseases, number				<0.001				0.580
0	118 (41.4)	102 (35.8)	65 (22.8)		76 (26.9)	116 (41.0)	91 (32.2)	
1	53 (30.3)	80 (45.7)	42 (24.0)		39 (22.3)	68 (38.9)	68 (38.9)	
>=2	25 (18.0)	90 (64.7)	24 (17.3)		34 (24.5)	60 (43.2)	45 (32.4)	
Chronic therapy				<0.001				0.491
No	127 (41.6)	107 (35.1)	71 (23.3)		81 (26.6)	125 (41.1)	98 (32.2)	
Yes	65 (22.7)	163 (57.0)	58 (20.3)		65 (22.8)	118 (41.4)	102 (35.8)	

Legend: COPD, chronic obstructive pulmonary disease; ESRD, end stage renal disease.

**Table 3 vaccines-09-00172-t003:** Clinical characteristics of patients.

	Influenza Vaccine 2020				SARS-CoV-2 Vaccine 2020			
	Unlikely	Likely	Undecided	*p*-Value	Unlikely	Likely	Undecided	*p*-Value
	*n* (%)	*n* (%)	*n* (%)		*n* (%)	*n* (%)	*n* (%)	
COVID-19 status				0.495				0.110
Confirmed	192 (33.1)	261 (45.0)	127 (21.9)		146 (25.3)	239 (41.3)	193 (33.4)	
Suspected	4 (21.0)	11 (57.9)	4 (21.0)		3 (15.8)	5 (26.3)	11 (57.9)	
Symptomatic COVID-19				0.313				0.153
No	13 (23.6)	29 (52.7)	13 (23.6)		8 (14.5)	28 (50.9)	19 (34.5)	
Yes	182 (33.6)	242 (44.7)	117 (21.6)		141 (26.2)	213 (39.5)	185 (34.2)	
Duration of symptoms, days				0.144				0.273
<7	13 (35.1)	18 (48.6)	6 (16.2)		6 (16.2)	20 (54.0)	11 (29.7)	
7–13	23 (47.9)	15 (31.2)	10 (20.8)		14 (29.2)	16 (33.3)	18 (37.5)	
14–20	18 (30.5)	21 (35.6)	20 (33.9)		13 (22.4)	17 (29.3)	28 (48.3)	
21–30	17 (27.9)	31 (50.8)	13 (21.3)		15 (24.6)	24 (39.3)	22 (36.1)	
31–60	43 (30.9)	65 (46.7)	31 (22.3)		30 (21.7)	59 (42.7)	49 (35.5)	
>60	34 (43.0)	28 (35.4)	17 (21.5)		26 (32.9)	28 (35.4)	25 (31.6)	
Hospital admission				0.001				0.800
No	160 (36.2)	178 (40.3)	104 (23.5)		111 (25.2)	174 (39.5)	155 (35.2)	
Hospital ward	32 (23.9)	78 (58.2)	24 (17.9)		32 (23.9)	61 (45.5)	41 (30.6)	
ICU	4 (17.4)	16 (69.6)	3 (13.0)		6 (26.1)	9 (39.1)	8 (34.8)	
Length of stay, days				0.723				0.336
0–3	6 (20.0)	18 (60.0)	6 (20.0)		6 (20.0)	15 (50.0)	9 (30.0)	
4–7	8 (25.8)	19 (61.3)	4 (12.9)		5 (16.1)	17 (54.8)	9 (29.0)	
8–14	9 (28.1)	20 (62.5)	3 (9.4)		12 (37.5)	16 (50.0)	4 (12.5)	
15–30	4 (28.6)	10 (71.4)	0 (0.0)		5 (35.7)	7 (50.0)	2 (14.3)	
>30	0 (0.0)	4 (100)	0 (0.0)		1 (25.0)	1 (25.0)	2 (50.0)	
Severity classification				0.086				0.486
Asymptomatic	13 (23.6)	29 (52.7)	13 (23.6)		8 (14.5)	28 (50.9)	19 (34.5)	
Mild disease	151 (36.9)	162 (39.6)	96 (23.5)		106 (26.0)	158 (38.8)	143 (35.1)	
Moderate disease	22 (23.7)	55 (59.1)	16 (17.2)		26 (28.0)	39 (41.9)	28 (30.1)	
Severe disease	6 (25.0)	15 (62.5)	3 (12.5)		4 (16.7)	11 (45.8)	9 (37.5)	
Critical illness	3 (20.0)	10 (66.7)	2 (13.3)		5 (33.3)	5 (33.3)	5 (33.3)	
Recovering signs and feeling				0.551				0.272
No	39 (29.6)	65 (49.2)	28 (21.2)		40 (30.3)	53 (40.1)	39 (29.5)	
Yes	151 (33.6)	198 (44.0)	101 (22.4)		107 (23.8)	185 (41.2)	157 (35.0)	
Symptoms of persistence				0.580				0.063
No	122 (34.1)	162 (45.2)	74 (20.7)		77 (21.6)	149 (41.8)	130 (36.5)	
Yes	74 (30.7)	110 (45.6)	57 (23.6)		72 (29.9)	95 (39.4)	74 (30.7)	

Legend: ICU, intensive care unit.

**Table 4 vaccines-09-00172-t004:** Reasons for vaccine hesitancy (*n* = 90).

Reasons for Vaccine Hesitancy	*n* (%)
I am concerned about the safety and/or the side effects	21 (14.1)
I am concerned because I don't think the vaccine will be effective	28 (18.8)
I don’t think I will need the vaccine due to previous infection, health status or age	31 (20.8)
I am against vaccines in general	18 (12.1)
I can’t take any vaccine because of previous vaccine reactions	2 (1.3)
I don’t know	49 (32.9)

## Data Availability

The data presented in this study are available on request from the corresponding author (M.P.). The data are not publicly available due to privacy concerns.

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
