# Peer review of "Vaccine Hesitancy among Italian Patients Recovered from COVID-19 Infection towards Influenza and Sars-Cov-2 Vaccination"

_vaccines, 2021, doi:10.3390/vaccines9020172_

Round 1

Reviewer 1 Report

This is a timely and interesting manuscript on vaccine hesitancy among patients in Italy who have recovered from COVID-19 infection. Overall, the manuscript is good, however it can be improved by narrowing the focus of the paper.

General comments:

Why were only COVID-19 patients sampled, what was the hypothesis behind this? That they would be more likely to be vaccinated/or less likely? It would be good to include the context/justification as to why this group was chosen. And why people who had not been infected were not included - they may be less hesitant than people who think they have immunity? Limitations of the study design should be included in the discussion.

More specific comments:

Introduction

You state in the introduction that non-pharmaceutical interventions seem to be insufficient. This is not correct. Countries who closed their borders and implemented strict public health measures either have had no COVID-19 cases or have eliminated SARS-CoV-2. 

You could add in some context to the number of people who had flu in this region in 2019 and 2020. Have rates dropped due to public health COVID-19 measures as they have elsewhere in the world? 

Materials and methods

You state that you carried out the study at the Academic Hospital Udine. This needs to be re-worded and corrected as it is not a hospital-based study as this implies. Rather participants were either previous inpatients or outpatients and were contacted by phone.

How were the participants selected? Was is all patients >18 years? What were the inclusion/exclusion criteria?  Where there quotas to try and get a range of ages, ethnicities, urban/rural patients etc?

Your methods mention very little about variables reported in your tables, occupation, nationality, BMI, smoking habits, alcohol consumption and clinical characteristics. How were these defined and analysed? Where did the data come from?

Results

I am unsure why you have included so many variables in Table 1 - what is the hypothesis behind including these variables? Do you expect people with high/low BMIs to be more likely not to be vaccinated? I would take out all variables that are not related to the aim of your paper.

Table 2 and 3 could be combined and clinical characteristics that are not related to the aim of the study deleted. The two tables seem to be a bit of a double up. The numbers for anxiety, depression, cardiovascular disease, ESRD, liver disease are all very low, I would delete them. Severity of disease, length of stay, duration of symptoms, hospitalisation are more meaningful to the aim of your paper.  

What does home medication mean? 

Table 4 – a lot of participants gave no answer as to why they wouldn't have a vaccine, this needs to be mentioned in the discussion as a limitation.

Discussion:

The discussion could be shorter, please delete any repeated material. I would focus on the findings around COVID-19 vaccination, rather than flu. What would be interesting to discuss would be what can potentially be done to encourage people (who have previously had COVID-19) to be vaccinated, given you have found that more than half are hesitant or undecided. If people are not going to be vaccinated (at levels to achieve herd-immunity) then what can governments do to curb the COVID-19 pandemic.

Author Response

This is a timely and interesting manuscript on vaccine hesitancy among patients in Italy who have recovered from COVID-19 infection. Overall, the manuscript is good, however it can be improved by narrowing the focus of the paper.
General comments:
Why were only COVID-19 patients sampled, what was the hypothesis behind this? That they would be more likely to be vaccinated/or less likely? It would be good to include the context/justification as to why this group was chosen. And why people who had not been infected were not included - they may be less hesitant than people who think they have immunity? Limitations of the study design should be included in the discussion.

Dear reviewer,

we really appreciated your feedback and addressed all suggestions as required.

The Reviewer is right to point out the interest of adding data on the context of study population.  In our study, we decided to focus on COVID-19 patients because they may have an ambivalent attitude, since one the one hand experience of previous critical illness could be a determinant for vaccine attitude but on the other hand, perception of protection due to immunization could condition vaccine tendency.
We agree that absence of a control group composed of individuals without previous SARS-CoV-2 infection is an important limitation, since vaccine attitude infection might be associated also with lockdown or changes in lifestyle habits, pandemic insecurity, work loss and poverty. However, our findings are in line with recent surveys performed on general population in Italy (Caserotti et al. Soc Sci Med. 2021 Jan 7). Justification of the study population and limitations of the study design have been included in the discussion. We have added these sentences in in the discussion page 10, lines 210-215 and page 12, lines 297-306.

More specific comments:

Introduction
You state in the introduction that non-pharmaceutical interventions seem to be insufficient. This is not correct. Countries who closed their borders and implemented strict public health measures either have had no COVID-19 cases or have eliminated SARS-CoV-2. 

REPLY: The referee is right to point out to clarify this issue on non-pharmaceutical interventions (NPI). We agree that as long as there is no effective and safe vaccine to protect those at risk of severe COVID-19, NPI are the most effective public health interventions against COVID-19 and can significantly contain the COVID-19 pandemic (https://www.ecdc.europa.eu/en/publications-data/covid-19-guidelines-non-pharmaceutical-interventions).

Governments worldwide have implemented varying degrees of restrictions on population movement to slow the spread of SARS COV2 but with different epidemiological experiences and responses (Han et al. Lancet 2020; 396: 1525–34). However, in our region (Friuli Venezia Giulia, Udine), NPI were insufficient to control the spread of SARS COV 2 (https://covid19map.protezionecivile.fvg.it/)

We have added some comments in the introduction page 1, lines 34-42

You could add in some context to the number of people who had flu in this region in 2019 and 2020. Have rates dropped due to public health COVID-19 measures as they have elsewhere in the world? 

REPLY::We agree with the referee that this is an interesting topic. The convergence of a simultaneous seasonal influenza epidemic was of great concern at the beginning of the 2020-2021 winter season. However, measures meant to control SARS COV2 pandemic have shown wide-ranging implications also on influenza and most other respiratory diseases in the Southern and Northern  Hemisphere (Yeoh et al Clin Infect Dis 2020 Sep 28;ciaa1475;https://www.epicentro.iss.it/influenza/flunews).  We have added some comments and bibliography on evolution of flu season in our region in the introduction page 2, lines 40-42

Materials and methods
You state that you carried out the study at the Academic Hospital Udine. This needs to be re-worded and corrected as it is not a hospital-based study as this implies. Rather participants were either previous inpatients or outpatients and were contacted by phone. How were the participants selected? Was is all patients >18 years? What were the inclusion/exclusion criteria?  Where there quotas to try and get a range of ages, ethnicities, urban/rural patients etc?Your methods mention very little about variables reported in your tables, occupation, nationality, BMI, smoking habits, alcohol consumption and clinical characteristics. How were these defined and analysed? Where did the data come from?

REPLY: The referee is right to point out to clarify Material and methods paragraph. For this cohort study, we approached the General Hospital and Microbiology of Academic Hospital Udine, Italy, which serves all 530 000 inhabitants of Udine province. In our context, swabs were performed in all hospitalized and non-hospitalized patients visiting the infectious disease department. The target population were eligible patients > 18 years, as suspected or confirmed COVID-19 cases, and willing to participate in a telephonic interview performed after 6 months of the COVID-19 onset. There was no filter selection. As reported in the result section, there were excluded patients who refused to participate in the research, those who were living in nursing home or long-term facilities and not capable to answer due to cognitive decline, as well as those lost-to-follow-up and died.

Patients willing to participate were included and a database including their demographic, clinical and laboratory data, was populated. Moreover, patients were telephone interviewed by trained nurses after 6 months of the COVID-19 onset.

We examined factors that have been associated with vaccine hesitancy in previous studies: age, sex, country of birth, baseline comorbidities, alcoholism, smoking habit, work, personal experiences with vaccinations (https://www.who.int/news-room/spotlight/ten-threats-to-global-health-in-2019). We have added these sentences in an appropriate manner in the Material and Methods pages 3, lines 99-101.

Moreover we have also acknowledged among the limitations that some variables – as ethnicity and place of living have not been investigated. Thank you.

Results
I am unsure why you have included so many variables in Table 1 - what is the hypothesis behind including these variables? Do you expect people with high/low BMIs to be more likely not to be vaccinated? I would take out all variables that are not related to the aim of your paper. Table 2 and 3 could be combined and clinical characteristics that are not related to the aim of the study deleted. The two tables seem to be a bit of a double up. The numbers for anxiety, depression, cardiovascular disease, ESRD, liver disease are all very low, I would delete them. Severity of disease, length of stay, duration of symptoms, hospitalisation are more meaningful to the aim of your paper.  

REPLY: As previously suggested, we examined factors that have been associated with vaccine hesitancy in previous studies: age,  gender, country of birth, baseline comorbidities, alcoholism, smoking habit, work, personal experiences with vaccinations (https://www.who.int/news-room/spotlight/ten-threats-to-global-health-in-2019).  Moreover, we have removed several aspects – as you suggested – not consistent with the literature available. Therefore, in accordance with the reviewer suggestions, we have now changed Table 1 and Table 2 and we have added Table A supplemental

What does home medication mean? 
REPLY:  It means the chronic therapy. We have substituted the word 

Table 4 – a lot of participants gave no answer as to why they wouldn't have a vaccine, this needs to be mentioned in the discussion as a limitation.

REPLY:  In keeping with the reviewer suggestion, we believe that among hesitant patients towards SARS-CoV-2 vaccine, the majority reported no arguments and this might be a limitation but also reflect the uncertainty generated by the COVID-19 outbreak. As requested, we have added comments in the discussion.

Discussion:
The discussion could be shorter, please delete any repeated material. I would focus on the findings around COVID-19 vaccination, rather than flu. What would be interesting to discuss would be what can potentially be done to encourage people (who have previously had COVID-19) to be vaccinated, given you have found that more than half are hesitant or undecided. If people are not going to be vaccinated (at levels to achieve herd-immunity) then what can governments do to curb the COVID-19 pandemic.

REPLY: In accordance with the reviewer suggestions, we have now reduced discussion and we have added some comments on measures for building confidence in COVID 19 vaccine in hesitant or undecided people. We have in depth revised the discussions. In light of the suggestions, we have also revised the conclusion by providing insights regarding the actions.

Reviewer 2 Report

In this paper, the authors explored the  COVID-19 patient’s tendency towards the flu vaccine and a putative COVID-19 vaccine. The secondary objective is the identification of associated factors of hesitancy. They interviewed 599 patients who had been infected with SARS-CoV-2 during the first pandemic wave (from March to May 2020).

In general, the study is well designed and well presented, the discussion is very well argumented.

I have some minor concerns:

- There are many groupings in table 1 and 2 and it is difficult to folllow the data. As suggestion, to simplify the reading of the data, Table 1 and 2 can report in the first column the Likely % and in the second the Unlikely. 

- In the results section, the authors indicated the percentage and the absolute numebers of patients for each data. This makes the paper difficult to read. I think that only the percentage should be indicated while the number is reported in the table. Pay attention to a wrong data in line 134.

- In the abstract, the numerical data should be avoided anche only the significance of data should be reported.

Author Response

REVIEWER 2

In this paper, the authors explored the COVID-19 patient’s tendency towards the flu vaccine and a putative COVID-19 vaccine. The secondary objective is the identification of associated factors of hesitancy. They interviewed 599 patients who had been infected with SARS-CoV-2 during the first pandemic wave (from March to May 2020).
In general, the study is well designed and well presented, the discussion is very well argument.

Dear reviewer

We thank you for the valuable feedback and the appreciation of our manuscript. We have considered all suggestions as reported below. 

I have some minor concerns: - There are many groupings in table 1 and 2 and it is difficult to follow the data. As suggestion, to simplify the reading of the data, Table 1 and 2 can report in the first column the Likely % and in the second the Unlikely. 
REPLY: we have considered with care this suggestion and provided some changes in the Table 1 and 2 in order to render them more approachable for the readers. Moreover, in keeping with the reviewer’s recommendation and the Vaccines journal formatting requirements and style we have now changed Table 1 and Table 2. We thank you for your suggestions.

In the results section, the authors indicated the percentage and the absolute numbers of patients for each data. This makes the paper difficult to read. I think that only the percentage should be indicated while the number is reported in the table.

REPLY: We have also considered with care this suggestion. We have removed – as you can see – some redundancies both in the text and in the table. However, with the intent to follow the Vaccines journal formatting requirements and style as suggested by the Editor, we have left some percentages and frequencies. In summary, we have changed Table 1 and Table 2 and some specific points in the body of the text.

We thank you for your suggestions.

Pay attention to a wrong data in line 134. 

REPLY: Corrected, we apologise for the mistake- thank you.

 In the abstract, the numerical data should be avoided anche only the significance of data should be reported.

REPLY: we have considered with care this suggestion. We have slightly changed the abstract according to your recommendations. However, we have also considered the Vaccines journal formatting requirements and style as recommended by the Editor. 

Round 2

Reviewer 1 Report

Thank you for clarifying my concerns and addressing them. I am happy with the changes the authors have made.